# Effect of Mo Content on In-Situ Anisometric Grains Growth and Mechanical Properties of Mo_2_FeB_2_-Based Cermets

**DOI:** 10.3390/ma15196729

**Published:** 2022-09-28

**Authors:** Yupeng Shen, Wuxi Xie, Bingbing Sun, Yunfei Liu, Yajin Li, Zhen Cao, Yongxin Jian, Zhifu Huang

**Affiliations:** 1Xi’an Modern Chemistry Research Institute, Xi’an 710065, China; 2State Key Laboratory for Mechanical Behavior of Materials, Xi’an Jiaotong University, Xi’an 710049, China

**Keywords:** Mo_2_FeB_2_-based cermets, Mo addition, microstructure, mechanical properties

## Abstract

Mo_2_FeB_2_-based cermets have wide applications in fields of wear resistance, corrosion resistance and heat resistance due to their simple preparation process, low-cost raw materials, and prominent mechanical properties. Herein, Mo_2_FeB_2_-based cermets with *x*Mo (*x* = 43.5, 45.5, 47.5, 49.5, wt.%) were prepared by means of the vacuum liquid phase sintering technique. Investigations on the microstructure and mechanical properties of Mo_2_FeB_2_-based cermets with Mo addition were performed. Experimental results show that, with Mo content increasing, the average particle size decreases gradually, revealing that the grain coarsening of Mo_2_FeB_2_-based cermets is controlled by interface reaction. In addition, Mo_2_FeB_2_ grains gradually transform from an elongated shape to a nearly equiaxed shape. The improvement of Mo_2_FeB_2_ hard phase on the morphology is mainly due to the inhibition of solution–precipitation reaction by increasing Mo. Furthermore, the relative density of cermets decreases due to the reduced Fe content. When Mo content is 47.5 wt.%, a relatively small grain size of Mo_2_FeB_2_ is obtained (about 2.03 μm). Moreover, with the increase in Mo content, hardness and transverse rupture strength (TRS) of Mo_2_FeB_2_-based cermets increase firstly and then decrease. Whereas, with increasing Mo content, the fracture toughness deteriorates gradually. When Mo content is 47.5 wt.%, the comprehensive mechanical properties of cermets are the best. The optimal raw material ratio for the preparation of Mo_2_FeB_2_-based cermets in this study is determined to be 47.5 wt.% Mo–6.0 wt.% B-Fe.

## 1. Introduction

Borides show good hardness and good chemical stability [1]. Hence, borides are potential antiwear materials and have been widely investigated by many researchers [2,3]. However, the further application of borides in industry is halted owing to their intrinsic brittleness and poor sinterability. Reaction boronizing sintering is a novel strategy, wherein binary borides react with metals to form ternary borides [4,5]. Through reaction boronizing sintering strategy, Mo_2_FeB_2_-based cermets, Mo_2_NiB_2_-based cermets and WCoB-based cermets have been successfully developed [5,6,7,8,9,10,11,12]. Ternary boride-based cermets have the advantages of metal and ceramic [13]. Thus, they exhibit high hardness, high transverse rupture strength, low density, good wear resistance and corrosion resistance. Therein, Mo_2_FeB_2_ based cermets receive a lot of attention for their cheap raw materials and excellent overall performance [14]. Mo_2_FeB_2_ based cermets have been widely used in molds and cutting tools [15,16].

At present, many research groups have conducted research on Mo_2_FeB_2_ based cermets, investigating the microstructure evolution and mechanical properties of cermets with doped elements, such as Cr, V, Mn, Ti and C [17,18,19,20,21,22,23,24]. There are also some research studies on the preparation of cermets [25,26,27]. Mo_2_FeB_2_ based cermets are prepared through a powder metallurgy process which includes the steps of batching, mixing, compacting, and sintering. Therein, batching is the first step in preparing Mo_2_FeB_2_ based cermets, in which two parameters need to be designed. The first parameter is the Mo/B atomic ratio. Tsuneyuki and Yu reported that, when the Mo/B atomic ratio was 0.9, the cermets showed fine particles, homogeneous two-phase microstructure and the highest TRS [28,29]. Thus, the Mo/B atomic ratio of 0.9 is widely adopted. The second parameter that needs to be considered is the content of Mo, which determines the relative ratio of Mo_2_FeB_2_ hard phase and Fe binder phase. When Mo/B atomic ratio of 0.9 is fixed, the relative content of Mo_2_FeB_2_ hard phase will be high if Mo content is high. There is no doubt that the microstructure and mechanical properties of cermets are affected by the content ratio of hard phase and bind phase. However, the influence of the content ratio of Mo_2_FeB_2_ hard phase and Fe binder phase on cermets is still unknown. Even the Mo contents used by different research studies are different. For example, the Mo content used in Wu’s studies is 44.4 wt.% [8], the Mo content used in Yu’s studies is 48.0 wt.% [26], and the Mo content used in Ivanov’s studies is 55.0 wt.% [6]. Therefore, the investigation concerning the effects of Mo on the microstructure evolution and properties of Mo_2_FeB_2_ based cermets has great theoretical value.

Herein, four series of Mo_2_FeB_2_ based cermets with *x*Mo (*x* = 43.5, 45.5, 47.5, 49.5, wt.%) contents were prepared. The effects of Mo on the microstructure and mechanical properties of Mo_2_FeB_2_ based cermets are studied. 

## 2. Materials and Methods

### 2.1. Raw Materials and Composition Design

Raw materials used in this study include Mo, FeB, and carbonyl Fe, shown in Figure 1. As shown, Mo powder has a hammer ball morphology, and the morphology of Fe powder is spherical, while the FeB powder has an irregular morphology. Table 1 shows the characteristics of raw powders [23]. As shown, the Mo/B atomic ratio of 0.9 is fixed and Mo content is changed. The nominal compositions of Mo_2_FeB_2_ based cermets used in the present research are presented in Table 2. 

### 2.2. Fabrication Process

The Mo, FeB and carbonyl Fe were weighed as per the proportion presented in Table 2, and then milled in stainless steel tanks. After that, the slurries were dried, pressed into green compacts, and then sintered. Figure 2 displays the sintering curve. The more detailed preparation process was reported in our previous works [25,27].

### 2.3. Characterization

The phase analysis was performed by X-ray diffraction (XRD, D/Max-2400). Scanning electron microscopy (SEM, VegaII XMU) was used to observe the microstructure of cermets and bending fracture morphology at 20 kV. According to stereology, the volume fraction of Mo_2_FeB_2_ hard phase in cermets was calculated using 20 different SEM micrographs via Image J software (Image J 1.8). Image analysis software (Image Pro Plus, Version 6.0) was used to measure the particle size and shape factor of Mo_2_FeB_2_ in cermets. Equivalent circle method was used to measure the particle size of Mo_2_FeB_2_ hard phase in cermets [30]. Particle size (D) was calculated via using Equation (1):(1)$$D=2\sqrt{\frac{S}{\pi }}$$
where S is the area of single particle.

Shape factor (K) was calculated via using Equation (2) [31,32]:(2)$$K=\frac{4\pi S}{L^{2}}$$
where S is the area of single particle and L is the perimeter of single particle. In order to ensure the reliability of test data, more than 500 particles were measured for each specimen. 

Relative density measurements of Mo_2_FeB_2_ based cermets were performed as per Archimedes technique [33]. Relative density (ρ) was calculated via using Equation (3):(3)$$\rho =\frac{m_{1}-m_{2}}{m_{3}-m_{2}}$$
where $m_{1}$, $m_{2}$ and $m_{3}$ are dry weight, suspending weight, and wet weight of specimen, respectively. 

Hardness of Mo_2_FeB_2_-based cermets at room temperature was determined under a hardness tester (HR-150A). Five samples with dimensions of 2.5 mm × 5 mm × 25 mm were prepared to test the fracture toughness (*K_IC_*) via single edge notched beam (SENB) method. The measurements were conducted using a three-point bend test. Fracture toughness (*K_IC_*) was calculated using Equation (4) [33].
(4)KIC=[1.93−3.07ah+14.53(ah)2−25.11(ah)3+25.80(ah)4]3Fl2bh2a12
where F is the critical rupture load, b is the width of specimen, l is the length of specimen, h is the thickness of specimen, and *a* is the prefabricated crack length.

Three-point bend test was used to measure the transverse rupture strength (TRS) of cermets. TRS (δ) was calculated by using Equation (5):(5)$$\delta =\frac{3Fl}{2bh^{2}}$$
where F is the critical rupture load, l is the length of specimen, b is the width of specimen, h is the thickness of specimen.

## 3. Results and Discussion

### 3.1. Microstructure Evolution

XRD patterns of Mo_2_FeB_2_ based cermets with different Mo contents are displayed in Figure 3. As shown, with Mo content increasing, the phases of cermets do not change. The primary phases are Mo_2_FeB_2_ hard phase and Fe binder phase, and there are no other phases. With the increase in Mo content from 43.5 wt.% to 49.5 wt.%, the degree of crystallinity of Mo_2_FeB_2_ decreases from 73.21% to 70.97% by analyzing the Jade software (Jade 6.0). With Mo content increasing, the intensity of diffraction peaks of Mo_2_FeB_2_ gradually increases, while the intensity of diffraction peaks of Fe decreases, indicating that the quantity of Mo_2_FeB_2_ increases gradually.

Figure 4 displays the microstructure of Mo_2_FeB_2_ based cermets with different Mo contents. As shown, the phase composition of cermets does not change with Mo content increasing. They are all composed of two phases of light and dark colors, namely, Mo_2_FeB_2_ hard phase and Fe binder phase, which is consistent with XRD results presented in Figure 3. When Mo content is from 43.5 wt.% to 47.5 wt.%, Mo_2_FeB_2_ particles are uniformly distributed in Fe binder phase. When the content of Mo is 49.5 wt.%, agglomeration of Mo_2_FeB_2_ particles occurs. In addition, with increasing Mo content, the relative content of Mo_2_FeB_2_ also gradually increases. For further quantitative analysis, the volume fraction of hard phase was statistically analyzed by image analysis software (Image Pro Plus 6.0). 

Figure 5 displays the volume fraction of hard phase with different Mo contents. With Mo content increasing from 43.5 wt.% to 49.5 wt.%, the volume fraction increases from 57.28% to 63.58%. This phenomenon mostly occurs because more Mo elements are added, which can generate more Mo_2_FeB_2_ hard phase in the solid phase reaction, thus increasing the volume fraction of Mo_2_FeB_2_ hard phase in cermets.

Meanwhile, with Mo content increasing from 43.5 wt.% to 49.5 wt.%, the particle size decreases and particles vary from an elongate shape to a nearly equiaxial shape. Furthermore, quantitative analyses of cermets were conducted via Image Pro Plus software (Image Pro Plus 6.0).

Figure 6 displays the statistical results of Mo_2_FeB_2_ particle size distribution with different Mo contents. As shown, with the increase in Mo content, the particle size distribution curve gradually moves to the left, that is, the lower limit, upper limit and peak value of particle size distribution range all decrease. When Mo content is 43.5 wt.%, the average particle size of Mo_2_FeB_2_ is 2.41 μm. There are a few particles, whose sizes range from 5.50 μm to 6.00 μm. This is due to excessive particle growth. When Mo content is 45.5 wt.% and 47.5 wt.%, the average particle size of Mo_2_FeB_2_ is 2.32 μm and 2.03 μm, respectively. When Mo content is 49.5 wt.%, the average particle size of Mo_2_FeB_2_ is 1.68 μm. However, there are a few particles, whose sizes range from 5.25 μm to 5.50 μm. This is due to the agglomeration of Mo_2_FeB_2_ hard phase resulting in excessive particle growth [26,27]. 

Results in Figure 5 and Figure 6 indicate that the average particle size of Mo_2_FeB_2_ decreases with increasing Mo content. The Mo_2_FeB_2_ particles grow up through the mode of solution–precipitation in the liquid matrix [29]. The growth rate is determined by both the solution–precipitation reaction rate at the solid–liquid interface and the diffusion rate of atoms in the liquid phase. Obviously, if the growth is diffusion controlled, the length of diffusion path between particles will affect the growth rate, and its rate will decrease with increasing liquid volume fraction, on the other hand, if it is interface reaction controlled, the rate will be independent, but may vary sensitively with the composition [30]. Thus, this result is the most direct proof that the grain coarsening of Mo_2_FeB_2_ based cermets is controlled by interface reaction. 

When the content of Mo increases, the content of Mo_2_FeB_2_ hard phase increases. Although the distance between Mo_2_FeB_2_ particles becomes shorter and the diffusion time decreases, the solution–precipitation reaction at solid–liquid interface of Mo_2_FeB_2_ particles is inhibited due to the decrease in the number of Fe-based liquid phase, resulting in the slow growth rate. The coarsening of particles takes more time to complete. Therefore, when Mo content is 43.5 wt.%, the hard phase content in cermets is relatively small, the liquid phase content is sufficient, the solution–precipitation reaction of Mo_2_FeB_2_ particles can be fully carried out, and the particles can be fully grown. When Mo content is from 45.5 wt.% to 47.5 wt.%, the hard phase content in cermets increases while the liquid phase content decreases. The solution–precipitation reaction of Mo_2_FeB_2_ particles is inhibited, and the particles cannot grow up completely, resulting in a small particle size. When Mo content increases to 49.5 wt. %, the hard phase content in cermet further increases and the liquid phase content decreases, and the solution–precipitation reaction of Mo_2_FeB_2_ hard phase particles is restrained, thus reducing the size of particles. However, the decrease in liquid content further shortens the distance between particles. The contact among particles intensifies, resulting in agglomeration of particles. Yu et al. have indicated that particle agglomeration would lead to the larger particles [26]. Therefore, hard phase agglomeration and formation of a small number of larger particles occurs.

Figure 7 displays the shape factor distribution of Mo_2_FeB_2_ hard phase with different Mo contents. With the increase in Mo content, the particle size distribution curve gradually moves to the right, that is, the lower limit, upper limit and peak value of particle size distribution range all increase. The average shape factor increases from 0.639 to 0.729, indicating that Mo_2_FeB_2_ particles gradually show obvious equiaxial characteristics with Mo content increasing. When Mo content is 43.5 wt.%, the liquid phase content is sufficient. Thus, the solution–precipitation reaction of Mo_2_FeB_2_ particles can be fully carried out, and the particles can be fully grown, showing obvious rod-like characteristics. When Mo content is from 45.5 wt.% to 49.5 wt.%, the liquid phase content decreases. The solution–precipitation reaction of Mo_2_FeB_2_ particles is inhibited. Therefore, particles cannot grow up completely, and gradually changed from long rod shape to nearly equiaxial shape. 

Figure 8 displays the relative density of Mo_2_FeB_2_-based cermets with different Mo contents. The relative density of cermets decreases monotonically with Mo content increasing. When Mo content is 43.5 wt.%, the relative density of cermets is 98.80%, and the relative density of cermets is 96.39% at Mo content of 49.5 wt.%. The decrease in relative density is mainly due to the decrease in liquid phase, which results in the inhibition of Mo_2_FeB_2_ particle rearrangement process in liquid phase to a certain extent. In addition, less liquid phase cannot adequately fill holes in cermets during sintering, which will also lead to a decrease in relative density. 

### 3.2. Mechanical Properties

Figure 9 shows the hardness of Mo_2_FeB_2_ based cermets with different Mo contents. As shown, with Mo content increasing, the hardness of the cermets increases first and then decreases, and the hardness reaches the maximum value of HRA 87.7 at Mo content of 47.5 wt.%. In this system, the factors that affect the hardness of cermets mainly include relative content of hard phase, the size of particles and the relative density of the cermets. With the increase in Mo content, the relative content of Mo_2_FeB_2_ increases, while average particle size decreases, which is conducive to the improvement of hardness, but the relative density of the cermets decreases gradually, which is bad for enhancing hardness. When Mo content is from 43.5 wt.% to 47.5 wt.%, the increase in hard phase content and the decrease in average particle size play a leading role, which leads to the increase in hardness value, and reaches the maximum value at Mo content of 47.5 wt.%. When Mo content is from 47.5 wt.% to 49.5 wt.%, the decrease in relative density plays a dominant role, which leads to the deterioration of hardness. 

Figure 10 shows the fracture toughness of Mo_2_FeB_2_ based cermets with different Mo contents. With the increase in Mo content, the fracture toughness decreases gradually from 15.9 MPa∙m^1/2^ to 12.0 MPa∙m^1/2^. In Mo_2_FeB_2_-based cermets, Mo_2_FeB_2_ hard phase is brittle, and toughness is mainly determined by binder phase. Therefore, the decrease in fracture toughness of cermets is mainly due to the decrease in binder phase content with increasing Mo content. Meanwhile, the decrease in relative density will increase the stress concentration and lead to the decrease in fracture toughness. In addition, with Mo content increasing, the particle size decreases, and the crack propagation path is shortened, which reduces the energy consumed during the crack propagation, resulting in the reduction in fracture toughness.

Figure 11 displays the TRS of cermets with different Mo contents. As shown, with Mo content increasing, TRS increases first and then decreases, and the cermets at Mo content of 47.5 wt.% obtain the maximum value of 1334.3 MPa. In this system, the TRS of cermets is mainly affected by particle size, shape of particles, content of binder phase and relative density of cermets. With increasing Mo content, the particle size decreases. According to the Hall–Petch formula, the fine particles are conducive to the improvement of TRS. The equiaxed particles can improve the deformation coordination between Mo_2_FeB_2_ and Fe, so that the cermets can be deformed uniformly during load bearing, which is conducive to the improvement of TRS. Moreover, the content of binder phase of cermets decreases with Mo content increasing, which will reduce the deformation of cermets in three-point bending tests and reduce its TRS. Furthermore, the reduction in relative density will increase the stress concentration and worsen TRS of cermets. Therefore, when Mo content is from 43.5 wt.% to 47.5 wt.%, the decrease in binder phase and relative density is not conducive to the improvement of TRS. However, the improvement of particle morphology plays a leading role in improving the TRS of cermets, and the maximum value of 1334.3 MPa is obtained at Mo content of 47.5 wt.%. When Mo content is from 47.5 wt.% to 49.5 wt.%, the stress concentration is intensified due to the obvious reduction in binder phase content, and the decrease in binder phase reduces the deformation of cermets, which plays a leading role in the deterioration of the TRS of cermets. 

Figure 12 shows a fractograph of the cermets after three-point bending tests with different Mo contents. The fracture mode of the cermets is a mixture of intergranular fracture and transgranular fracture. When Mo content is 43.5 wt.%, the fractograph shows the characteristics of an obvious transgranular fracture, which is mainly due to large particles. Meanwhile, more debonding can be observed. Mo_2_FeB_2_ particles grow up enough to cause the particles to appear as a long rod shape, which makes the deformation coordination between hard phase and binder phase worse. Thus, Mo_2_FeB_2_ particles are easy to separate from the Fe binder phase in three-point bending tests. When Mo content is 45.5 wt.%, the number of pores increases, while the number of transgranular fractures and interface debonding decreases. Therefore, the TRS of cermets increases slightly. When Mo content is 47.5 wt.%, more tearing edges are observed, which indicates that more cracks pass through the ductile Fe binder phase and consume more energy, thus improving the TRS of cermets. When Mo content is 49.5 wt.%, the number of pores increases obviously, which will increase the stress concentration and reduce TRS.

The microstructure and mechanical properties of Mo_2_FeB_2_-based ceramics with Mo content of 43.5 wt.%, 45.5 wt.%, 47.5 wt.% and 49.5 wt.% were investigated. The results show that when Mo content is 47.5 wt.%, Mo_2_FeB_2_ particles are fine and evenly distributed, and the comprehensive mechanical properties are the best. Therefore, the optimal raw material ratio for the preparation of Mo_2_FeB_2_-based cermets in this study is determined to be 47.5 wt.% Mo–6.0 wt.% B-Fe.

## 4. Conclusions

Herein, the effects of different Mo contents on in situ anisometric grains growth and mechanical properties of Mo_2_FeB_2_-based cermets were studied. The main conclusions are as follows: Grain coarsening of Mo_2_FeB_2_-based cermets is controlled by interface reaction.With Mo content increasing from 43.5 wt.% to 49.5 wt.%, Mo_2_FeB_2_ content increases while particle size and the relative density of the cermets decreases, and particles change from long rod shape to near-equiaxed shape. When Mo content is 47.5 wt.%, the cermets with dense microstructure, normal development of particles and uniform particle distribution can be obtained. In addition, the cermets with 47.5 wt.% Mo content exhibit a relatively small particle size of 2.03 μm.With Mo content increasing, hardness and TRS of cermets first increase and then decrease, while fracture toughness gradually decreases. When Mo content is 47.5 wt.%, hardness and TRS reach the maximum, being HRA 87.7 and 1334.3 Mpa, respectively. The comprehensive mechanical properties of cermets at 47.5 wt.% Mo content are the best.The optimal raw material ratio for the preparation of Mo_2_FeB_2_-based cermets in this study is determined to be 47.5 wt.% Mo–6.0 wt.% B-Fe. This study provides experimental basis for batching in preparing Mo_2_FeB_2_-based cermets, which is conducive to further application and promotion in industry.

## Figures and Tables

**Figure 1 materials-15-06729-f001:**
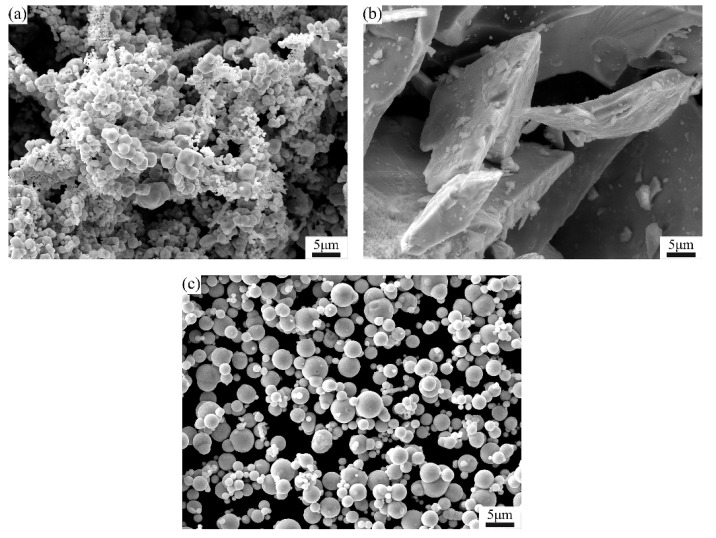
SEM micrographs of raw materials: (**a**) Mo; (**b**) FeB; (**c**) carbonyl Fe.

**Figure 2 materials-15-06729-f002:**
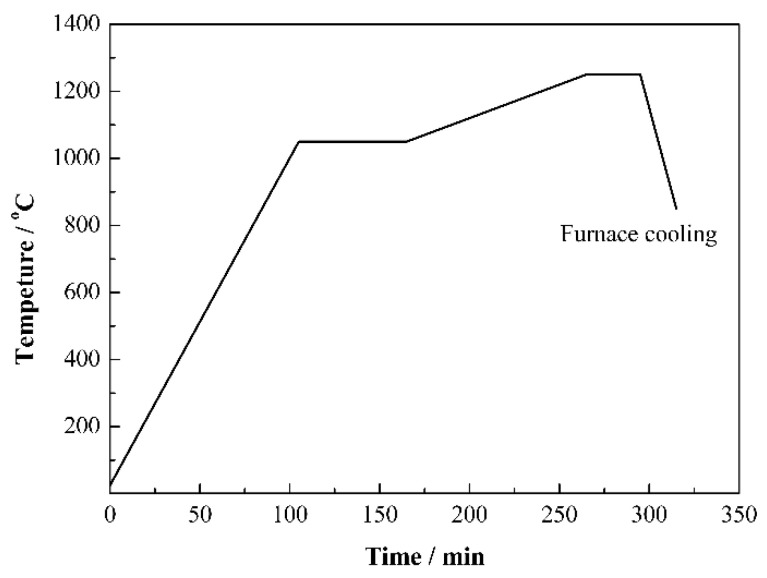
Sintering curve of Mo_2_FeB_2_ based cermets.

**Figure 3 materials-15-06729-f003:**
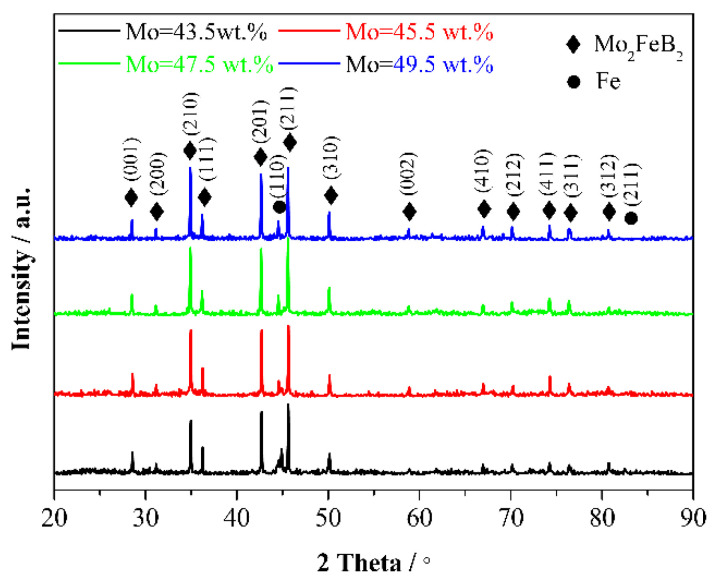
XRD patterns of Mo_2_FeB_2_ based cermets with different Mo contents.

**Figure 4 materials-15-06729-f004:**
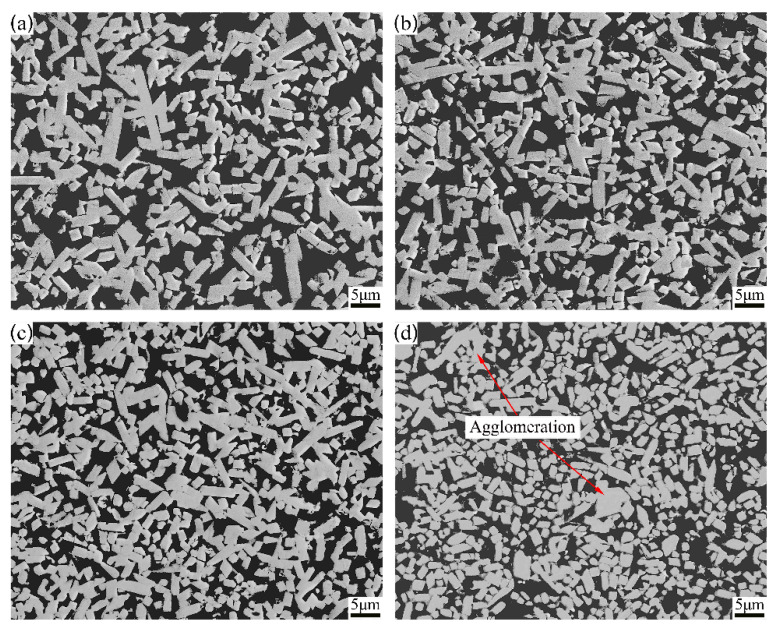
Microstructure of Mo_2_FeB_2_ based cermets with different Mo contents: (**a**) 43.5 wt.%; (**b**) 45.5 wt.%; (**c**) 47.5 wt.%; (**d**) 49.5 wt.%.

**Figure 5 materials-15-06729-f005:**
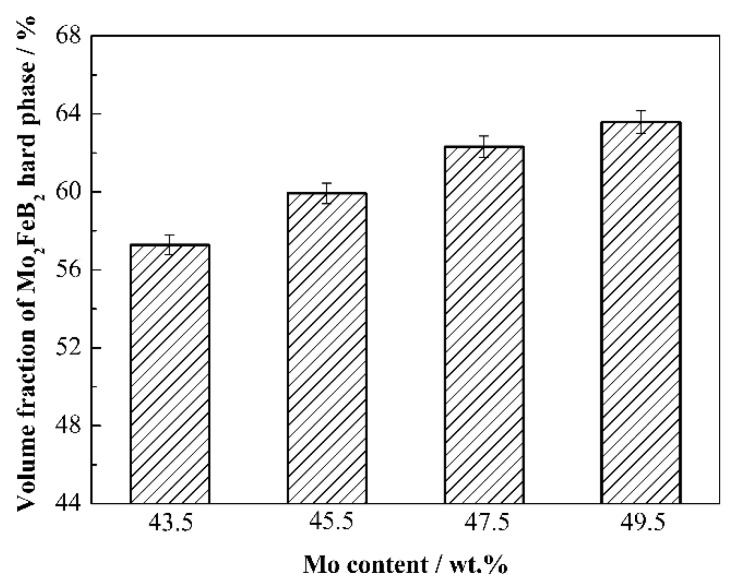
Volume fraction of Mo_2_FeB_2_ hard phase with different Mo contents.

**Figure 6 materials-15-06729-f006:**
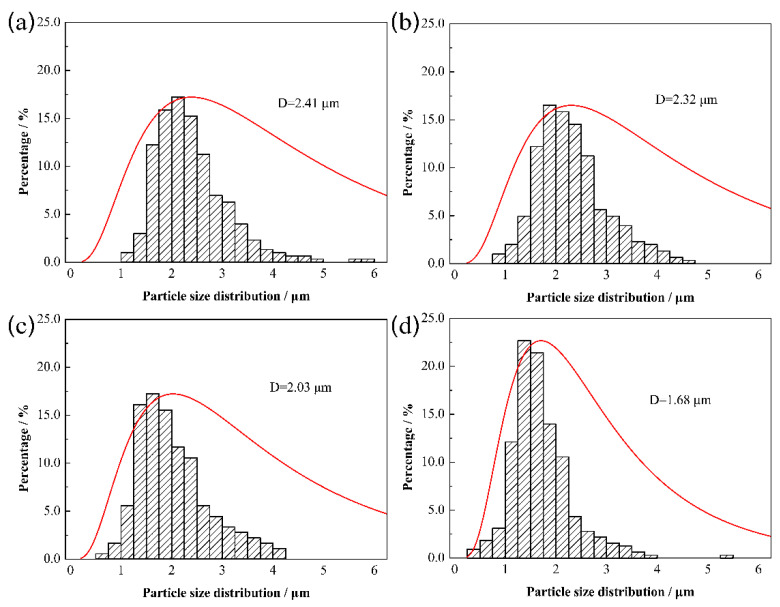
Particle size distribution of Mo_2_FeB_2_ with different Mo contents: (**a**) 43.5 wt.%; (**b**) 45.5 wt.%; (**c**) 47.5 wt.%; (**d**) 49.5 wt.%.

**Figure 7 materials-15-06729-f007:**
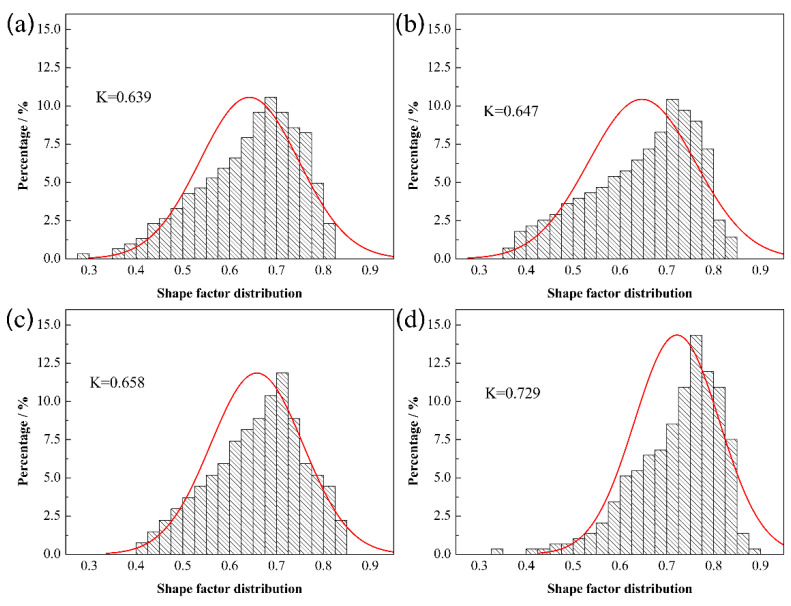
Shape factor distribution of Mo_2_FeB_2_ with different Mo contents: (**a**) 43.5 wt.%; (**b**) 45.5 wt.%; (**c**) 47.5 wt.%; (**d**) 49.5 wt.%.

**Figure 8 materials-15-06729-f008:**
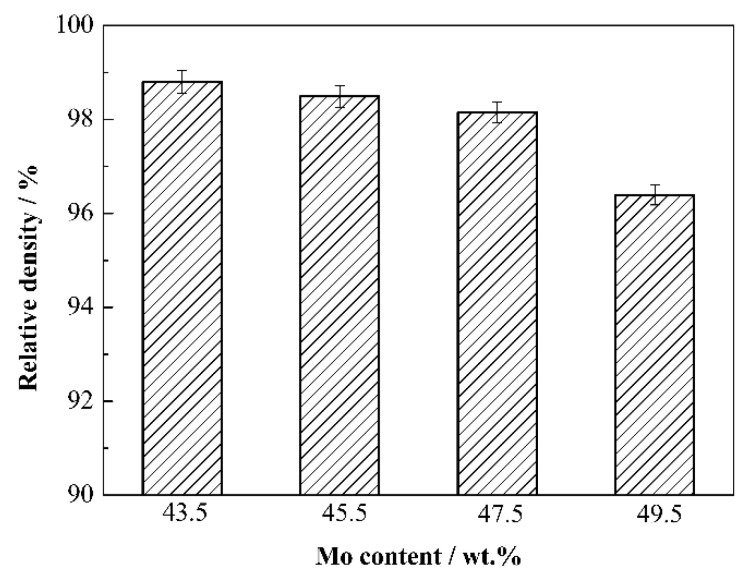
Relative density of Mo_2_FeB_2_-based cermets with different Mo contents.

**Figure 9 materials-15-06729-f009:**
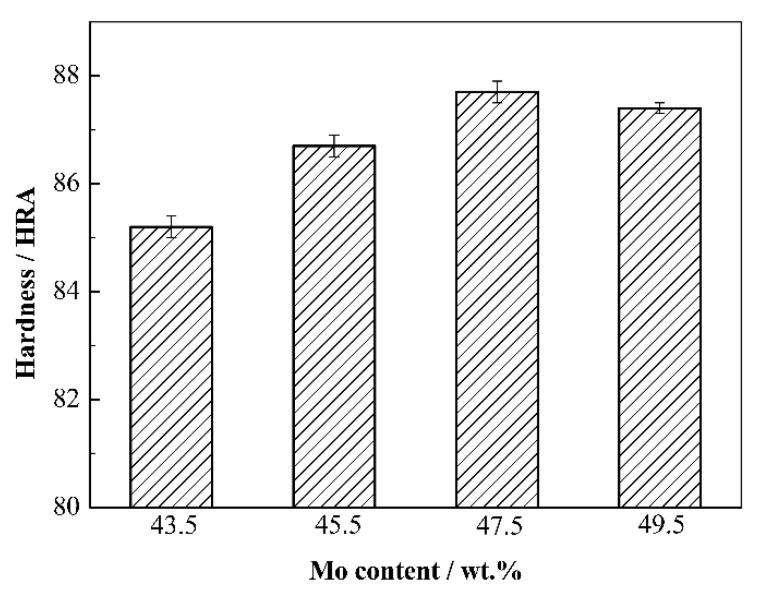
Hardness of Mo_2_FeB_2_ based cermets with different Mo contents.

**Figure 10 materials-15-06729-f010:**
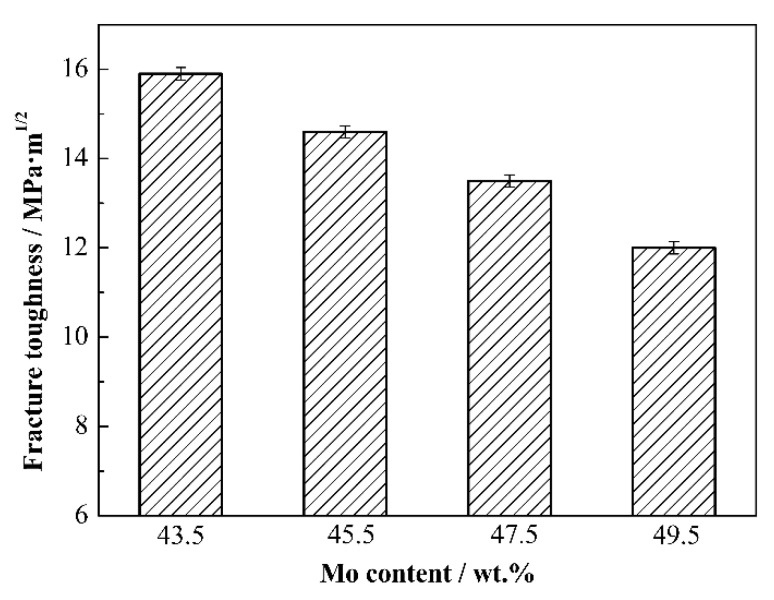
Fracture toughness of Mo_2_FeB_2_ based cermets with different Mo contents.

**Figure 11 materials-15-06729-f011:**
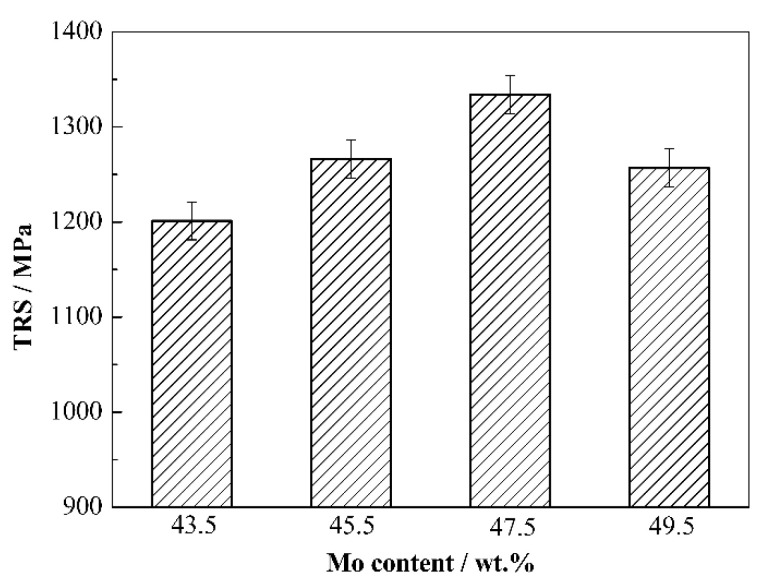
TRS of Mo_2_FeB_2_ based cermets with different Mo contents.

**Figure 12 materials-15-06729-f012:**
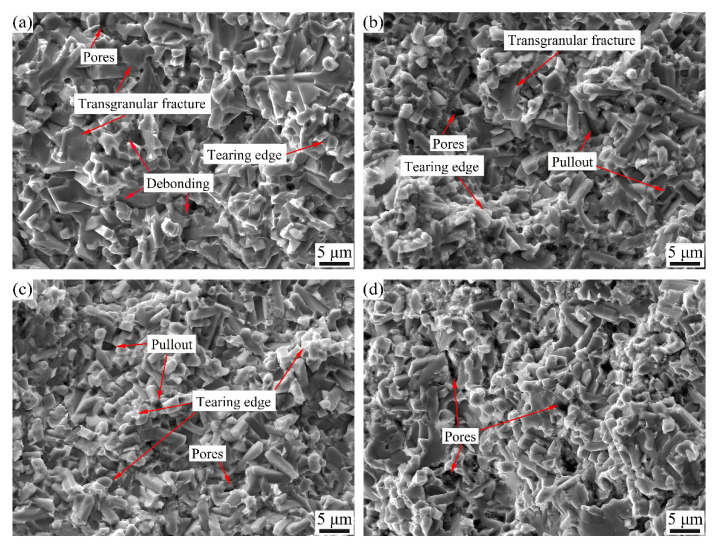
Fractograph of Mo_2_FeB_2_ based cermets after three-point bending tests with different Mo contents: (**a**) 43.5 wt.%; (**b**) 45.5 wt.%; (**c**) 47.5 wt.%; (**d**) 49.5 wt.%.

**Table 1 materials-15-06729-t001:** Characteristics of raw powders [23].

Powder	Mean Particle Size (μm)	Chemical Composition (wt.%)	Manufacturer
Mo	2	Mo ≥ 99.95, Fe < 0.005, Si < 0.002	Changsha Tianjiu Metal Material Corp., Ltd. of China
FeB	45	B = 20.05, Si < 0.36, C < 0.36	
Fe	5	Fe ≥ 99.81, C < 0.015, O < 0.16	

**Table 2 materials-15-06729-t002:** Compositions of Mo_2_FeB_2_-based cermets with different Mo content (wt.%).

Cermets	Mo	FeB	Fe
A	43.5	27.5	29.0
B	45.5	28.7	25.8
C	47.5	30.0	22.5
D	49.5	31.3	19.2

## Data Availability

Not Applicable.

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
