# Peer review of "Effect of Mo Content on In-Situ Anisometric Grains Growth and Mechanical Properties of Mo2FeB2-Based Cermets"

_materials, 2022, doi:10.3390/ma15196729_

Round 1

Reviewer 1 Report

In general, the manuscript is well presented and clear. To me, it was nice to read. A few real minor things... i) I would like to see which was the voltage condition were when performing SEM (in order to make it fully reproducible), ii) Check up at the parentheses in equation 5 and iii) there is a capital T on page 7 after reference 31: Thus, This... (lines are not numbered)

This is all from my side...

Author Response

Thank you very much for your time. The submitted manuscript “Effect of Mo content on in-situ anisometric grains growth and mechanical properties of Mo2FeB2 based cermets” (ID: materials-1920916) has been revised carefully according to your comments. All the mentioned issues have been addressed. And we have rechecked the full text to make sure no spelling and grammar mistakes. The revised part has been marked in red in the revised version. We hope the revised manuscript can meet the requirements for the publication on Materials.

The point-to-point responses are listed as follows:

Response to the reviewer:

Comments 1: I would like to see which was the voltage condition were when performing SEM (in order to make it fully reproducible).

Response 1: Thank you for your precious time and constructive advice. According to your suggestion, the voltage condition when performing SEM has been added in the revised manuscript on page 3.

Scanning electron microscopy (SEM, VegaII XMU) was used to observe the microstructure of cermets and bending fracture morphology at 20 kV.

Thank you very much again.

Comments 2: Check up at the parentheses in equation 5.

Response 2: Thank you for your precious time and constructive advice. According to your suggestion, the correct parentheses have been added in the revised manuscript on page 3. Thank you very much again.

Comments 3: There is a capital T on page 7 after reference 31: Thus, This... (lines are not numbered).

Response 3: Thank you for your constructive advice. The capital T after reference 31 has been corrected in the revised manuscript on page 6. Thank you very much again.

Reviewer 2 Report

In this study, reaction boronizing sintering was used to create Mo2FeB2 based cermets with various Mo concentrations. Investigated were the effects of various Mo contents on the development of the microstructure and the mechanical characteristics of Mo2FeB2 based cermets. These results merit for a publication and therefore without hesitation, I would recommend this article for a publication.

Author Response

Thank you very much.

Reviewer 3 Report

The article considers the effects of the addition of Mo on the evolution of microstructural properties and mechanical characteristics of Mo2FeB2 cermets. In general, the presented work is quite promising not only in terms of fundamental significance, but also in terms of practical applications of the results obtained for further research. The article corresponds to the subject of the declared journal, and the presentation of the material itself reflects in sufficient detail the essence of the results obtained. According to the reviewer, the article can be accepted for publication after the authors answer a number of questions that have arisen during its reading and analysis.

1. On the presented SEM images, the authors should provide explanations for the presence of agglomerates and the mechanisms responsible for their formation.

2. In the abstract, the authors should add one or two sentences about the prospects of the selected objects of Mo2FeB2 research, as well as their areas of application.

3. The results of X-ray phase analysis require additional explanation related to the evaluation of the phase composition, as well as the determination of the degree of crystallinity and phase ratio.

4. The authors should give an explanation in more detail about the mechanisms responsible for the change in strength characteristics with an increase in the concentration of Mo in the composition of Mo2FeB2, as well as the reasons for the decrease in strength properties at high concentrations of Mo.

5. In conclusion, it is necessary to indicate the future prospects of this study and its applicability for practical purposes.

Author Response

Thank you very much for your time. The submitted manuscript “Effect of Mo content on in-situ anisometric grains growth and mechanical properties of Mo2FeB2 based cermets” (ID: materials-1920916) has been revised carefully according to your comments. All the mentioned issues have been addressed. And we have rechecked the full text to make sure no spelling and grammar mistakes. The revised part has been marked in red in the revised version. We hope the revised manuscript can meet the requirements for the publication on Materials.

The point-to-point responses are listed as follows:

Response to the reviewer:

Comments 1: On the presented SEM images, the authors should provide explanations for the presence of agglomerates and the mechanisms responsible for their formation.

Response 1: Thank you for your precious time and constructive advice. The more detailed explanations for the presence of agglomerates and the mechanisms responsible for their formation has been provided in the revised manuscript on page 6.

The decrease of liquid content further shortens the distance between particles. The contact among particles intensifies, resulting in agglomeration of particles. Yu et al have indicated that particle agglomeration would lead to the larger partices [27].

Thank you very much again.

[27]   Yu, H. z.; Liu, W. j.; Zheng, Y., Microstructure and mechanical properties of liquid phase sintered Mo2FeB2 based cermets. Mater. Des. 2011, 32, (6), 3521-3525.

Comments 2: In the abstract, the authors should add one or two sentences about the prospects of the selected objects of Mo2FeB2 research, as well as their areas of application.

Response 2: Thank you for your precious time and constructive advice. According to your suggestion, the prospects of Mo2FeB2 based cermets, as well as their areas of application, have been added in the revised manuscript on page 1.

Mo2FeB2 based cermets have wide applications in fields of wear resistance, corrosion resistance and heat resistance due to their simple preparation process, low cost raw materials, and prominent mechanical properties.

Thank you very much again.

Comments 3: The results of X-ray phase analysis require additional explanation related to the evaluation of the phase composition, as well as the determination of the degree of crystallinity and phase ratio.

Response 3: Thank you for your constructive advice. The additional explanations have been added in the revised manuscript on page 4.

XRD patterns of Mo2FeB2 based cermets with different Mo contents were displayed in Fig. 3. As shown, with Mo content increasing, the phases of cermets don’t change. The primary phases are Mo2FeB2 hard phase and Fe binder phase. And there are no other phases. With the increase of Mo content from 43.5 wt.% to 49.5 wt.%, the degree of crystallinity of Mo2FeB2 decreases from 73.21% to 70.97% by analyzing the Jade software. Besides, with Mo content increasing, the intensity of diffraction peaks of Mo2FeB2 gradually increases, while the intensity of diffraction peaks of Fe decreases, indicating that the quantity of Mo2FeB2 increases gradually.

Thank you very much again.

Comments 4: The authors should give an explanation in more detail about the mechanisms responsible for the change in strength characteristics with an increase in the concentration of Mo in the composition of Mo2FeB2, as well as the reasons for the decrease in strength properties at high concentrations of Mo.

Response 4: Thank you for your precious time and constructive advice. According to your suggestion, the more detail explanations have been added in the revised manuscript on page 8.

Fig. 11 displays TRS of cermets with different Mo contents. As shown, with Mo content increasing, TRS increases first and then decreases. And the cermets at Mo content of 47.5 wt.% obtains the maximum value of 1334.3 MPa. In this system, TRS of cermets is mainly affected by particle size, shape of particles, content of binder phase and relative density of cermets. With increasing Mo content, the particle size decreases. According to Hall-Petch formula, the fine particle is conducive to the improvement of TRS. The equiaxed particles can improve the deformation coordination between Mo2FeB2 and Fe, so that the cermets can be deformed uniformly during load bearing, which is conducive to the improvement of TRS. While, the content of binder phase of cermets decreases with Mo content increasing, which will reduce the deformation of cermets in three-point bending tests and reduce its TRS. Meanwhile, the reduction of relative density will increase the stress concentration and worsen TRS of cermets. Therefore, when Mo content is from 43.5 wt.% to 47.5 wt.%, the decrease of binder phase and relative density is not conducive to the improvement of TRS. However, the improvement of particle morphology plays a leading role in improving the TRS of cermets. And the maximum value of 1334.3 MPa is obtained at Mo content of 47.5 wt.%. When Mo content is from 47.5 wt.% to 49.5 wt.%, the stress concentration is intensified due to the obvious reduction of binder phase content, and the decrease of binder phase reduces the deformation of cermets, which plays a leading role in the deterioration of TRS of cermets.

Thank you very much again.

Comments 5: In conclusion, it is necessary to indicate the future prospects of this study and its applicability for practical purposes.

Response 5: Thank you for your constructive advice. The future prospects of this study and its applicability for practical purposes have been added in the revised manuscript on page 10.

The optimal raw material ratio for the preparation of Mo2FeB2 based cermets in this study is determined to be 47.5 wt.% Mo-6.0 wt.% B-Fe. This study provides experimental basis for batching in preparing Mo2FeB2 based cermets, which is conducive to further application and promotion in industry.

Thank you very much again.

Round 2

Reviewer 3 Report

The authors answered all the questions, the article can be accepted for publication.